



# A decadal assessment of the climatology of aerosol and cloud properties over South Africa

Abdulaziz T. Yakubu[1] and Naven Chetty[1]

[1]Discipline of Physics, School of Chemistry and Physics, University of KwaZulu-Natal, Pietermaritzburg, 3209, South Africa

**Correspondence:** Naven Chetty (chettyn3@ukzn.ac.za)

**Abstract.** Aerosol-cloud interaction (ACI) plays an essential role in understanding precipitation occurrence and climate change but remains poorly understood. Conducting a climatology study on a regional or global scale constitutes a prospect better to understand ACI and its influence on precipitation and climate. This study analysed the characteristics of ACI over South Africa based on two instruments Moderate Resolution Imaging Spectroradiometer (MODIS) and Multiangle Imaging Spectroradiometer (MISR) onboard Terra satellite, and ground-based meteorology data from South Africa Weather Service (SAWS) between 2007-2016. The region mainly splits into the upper, central, and lower sub-regions based on the aerosol loading characteristics. Findings from the study show that depending on the atmospheric conditions, aerosol exhibits dual features of increasing and decreasing the potential formation of precipitating clouds . However, more often, fine-mode predominated aerosols suppress rain-bearing clouds. Furthermore, cloud top height (CTH) demonstrates upward increment from the lower to the upper part of the region, and the cloud fraction (CF) is in the downward direction. Both the CF and CTH display the characteristic enhancers of the precipitation intensity, mainly when the initial conditions necessary for rain-bearing occurs. Besides, cloud optical depth (COD) depends significantly on the liquid water path (LWP) and is suggestively associated with the aerosol-vapour ratio ingested into the cloud. Also notably, the temperature over the entire region generally increased steadily and continuously from 2013.

## 1 Introduction

The interaction between the atmospheric aerosol and cloud forms a vital driving force influencing climate change. Therefore, assessing aerosol and cloud formation over time is crucial to understanding climate variability on a regional and global scale (IPCC, 2013). Aerosols interact with solar radiation directly through scattering and absorption and indirectly with the cloud through cloud condensation nuclei (CCN) and ice nuclei (IN) formation to modify the earth's radiative budget (Ramanathan et al., 2001; IPCC, 2007). Through their interactions and solar radiation, aerosols and clouds influence changes in the hydrology cycle and climate pattern (Lohmann and Feichter, 2005; Kamala et al., 2009). Besides, they impact the reception and distribution of incoming solar radiation, thereby modifying the earth's energy budget (Haywood and Boucher, 2000). Various studies (e.g., Quaas et al., 2009; Jones and Christopher, 2010) attempt to quantify the effect of aerosol-cloud interaction on the earth's radiative forcing. However, the understanding remains challenging due to enormous associate uncertainty. To this end, understanding the climatology of aerosol and cloud over a region is vital to evaluating their roles on the earth's energy budget.



According to (IPCC, 2013), aerosol impact on climate mainly linked to their influence on radiative forcing through absorption and scattering of solar radiation represents the aerosol radiative interaction (ARI). Also, aerosols and clouds interaction that induces changes in the climate pattern by modifying the earth's energy budget and surface precipitation is the effective radiative forcing due to aerosol and cloud interaction (ERFaci). Thus, due to the complex and constant changes in aerosol and cloud

properties within the shortest time, the net effect on radiative forcing remains ambiguous and wrapped by huge uncertainty (IPCC, 2013). Besides, the task of unwrapping the complex link between the impact of aerosols and meteorology on cloud evolution constitutes a significant challenge towards understanding aerosol-cloud-climate interaction (Stevens and Feingold, 2009; Sorooshian et al., 2018). Since aerosols and clouds formation varies rapidly over a short time depending on the atmospheric emission sources, assessing the climatology over time can be highly useful for modelling and identifying potential impacts over

time and future projection. Therefore, a comprehensive assessment of aerosol and cloud climatology over time is crucial to understanding how climate change from small to significant scale impacts. The fundamental idea of the whole concept is to incorporate observation over a given long range of time to improve climate models on the impacts of aerosol-cloud interaction on climate change to enhance future variation prediction. Regional climate system offers a real advantage in understanding climate change holistically because of the distinct spectral and temporal differences in aerosol and cloud properties (Piketh et al., 2002;

Formenti et al., 2002) usher by the variation of anthropogenic influences over different regions. Besides, knowledge of regional climate coupled with global average will enhance the quality of general climate models. Furthermore, regional climate study will better understand how different weather and climate respond to varying atmospheric phenomena. For instance, previous studies have identified a high population of fine mode particles results in reduced droplet size under a constant liquid water path (Twomey, 1977) and precipitation suppression due to less coalescence (Albrecht, 1989). Similarly, other studies found a

large concentration of aerosol to invigorate cloud droplet concentration, increasing their vertical development and precipitation formation (Christensen and Stephens, 2012; Fan et al., 2016). Therefore, the impact of aerosol-climate interaction depends on multiple parameters that vary regionally due to distinct atmospheric and environmental conditions. For better quantification of ACI impacts on a global scale, detailed regional evaluation becomes a necessity. Although Africa mainly remains understudied in atmospheric and climate science, even though identified to be highly venerable to climate change impacts. Over

South Africa, a reasonable number of previous works have examined aerosol properties and, to some extent, cloud mostly over parts of the region. Studies, however, show that a considerable amount of aerosol dominates the northern part of the region. In contrast, low aerosol loading dominates the middle and lower part of the country (Formenti et al., 2002; Tesfaye et al., 2011). Nonetheless, the main discussion of many studies has focused on sources and the transport characteristics of aerosols. At the same time, their influence on the formation and evolution of clouds remains less investigated. Furthermore, the feedback effect

of cloud and precipitation on aerosol distribution, such as the effect of wet deposition on climatology, has not been quantified. Therefore, to significantly enhance our knowledge and improve climate models, the behaviour of aerosols over sites within the source proximity, the cloud response, and resulting climatic perturbation, more studies are needed around aerosol-cloud climatology. This work presents the effect of aerosol-cloud interaction on climate variability based on the climatology of aerosol and cloud over South Africa. This effort aimed towards improving our knowledge gap on the process of aerosol and cloud

formation and feedback along with proximity, including the climate impacts. Although climatology study is not holistic to



diagnose the challenge of climate change, it identifies the next level of research that will clarify things for better understanding. Besides, the result from such a study, once factored into the process of aerosol-cloud interaction, helps improve the general climate model. The study region is a suitable experimental environment to investigate aerosol-cloud interactions and how they influence the climate over proximity spatial domain. In this study, the region is divided in to three parts based on differences

in the aerosol loading as reported by previous study (Tesfaye et al., 2011) and as shown in Figure 1. The area generally shares the same climate pattern as the southern hemisphere regions. It possesses two distinct sub-climates divided along the upper, central, and lower parts of the region, a feature unique for the natural exploration of the effects of aerosol and cloud properties changes in the same season. Perhaps, a specific site like this will be suitable to understand the impact of seasonal changes on how aerosols are being transported or removed and the adverse effect on the cloud properties. Consequently, the climate system

over South Africa follows the characterisation by summer (December-February), fall (March-May), Winter (June-August), and spring (September-November) such that two periods of rainfall exist split along the upper, central, and lowest parts of the region. Hence, a duo biomass burning tenure occurs over the area. The upper and central parts experience summer rainfall and biomass burning during spring in preparation for the farming season. In contrast, the lower part receives rain during the winter and incidence of biomass burning during mid-summer till early fall. The influence of aerosols from one location on another is

critical to understanding climate variability over the region.

## 2    Data and methods

The datasets presented in this study are based on 10-years (2007-2015) measurements by two instruments, Multiangle Imaging Spectroradiometer (MISR) and Moderate Resolution Imaging Spectroradiometer (MODIS), onboard the Terra Earth Observing System (EOS) satellite. The satellite is a sun-synchronous orbiting system at an altitude of 705 Km descending north to south

poles and overpasses the equator about 10:30 h local time. The instruments aboard the satellite make a swarth dimension of 2330 Km to view the entire earth. In addition to the satellite data, collocated ground data for selected locations across the study region, acquired from the South Africa Weather Service (SAWS) is used to assess the meteorological impacts over the same period.

### 2.1    MISR data

MISR instrument launched more than two decades ago is equipped with nine cameras that view the earth at nine different angles; nadir, then 26.1, 45.6, 60.0, and 70.5 forward and backward of nadir above the planet to monitor the climate trends. With each camera, MISR acquires spatial sampling every 275 m and for about 7 minutes to cover a 360 Km swath view of the earth's surface (Diner et al., 1998; Kahn and Gaitley, 2015). Different datasets are produced from the instrument, ranging from raw observation to further processed data by subjecting to various screening operations. In this work, parameters from

the level-3 monthly aerosol product (MIL3MEA_4) grided at a resolution of 0.5 x 0.5 is used. Level-3 datasets represent a summary of level-2 data, so most uncertainties assessment and quality flags initiated in the level-2 algorithm processing are applicable. Retrieval of aerosol products by MISR is usually associated with various components that aggravate uncertain-





ties especially inherited from cloud contamination, sun glitter contamination, complex topographical terrain, shadows, particle property assumptions, and instrument calibration. To minimise and efficiently limit the uncertainties to aerosol measurement only, the observation data is passed through various quality flags algorithms for data pre-processing and screening operations (Kahn et al., 2009). Most importantly, cloud screening is initiated to eliminate cloud-contaminated data. Hence, only cloud-free data pixels are sampled using a cloud mask algorithm to avoid overestimation aerosol parameters. Further, the aerosol retrieval follows additional algorithm processing that employs different retrieval approaches for land and dark water (Kalashnikova et al., 2013; Witek et al., 2018) to get the columnal aerosol parameter that best characterise the region. This screening aims to eliminate data with glint contamination, which is most familiar with observations over the water and desert land. Implementation of these screening processes often reduces the percentage of the overall observation data available in the level-3 aerosol product, especially over cloudy regions, since only pixels that pass all tests are considered. However, several validation exercises on the aerosol product performed by coincident MISR and AERONET data have demonstrated significant agreement (e.g., Kahn et al., 2010; Martonchik et al., 2004; Abdou et al., 2005). Besides, several improvements in aerosol retrieval algorithms have been sought during the last decade to enhance aerosol product qualities over both land and dark water. Recent versions (e.g., V0022 and above) have all aerosol parameters at stage 2 validated, except for the aerosol optical depth (AOD) now at stage 3 validated. See https://asdc.larc.nasa.gov/documents/misr/misr_qual_stmts_current.html (last viewed: 27/02/22) for further information on quality statements and maturity level definitions. Subsequently, the MISR level-3 aerosol product is further equipped with additional quality flags retrieval masks that define realistic values of aerosol parameters over different regions to optimise usage. For the current study, datasets consisting of AOD, angstrom exponent (AE), and absorbing AOD ($A_{abs}$), together with aerosol index (AI) derived from the expression AOD * AE, are utilised to present part of the result.

## 2.2 MODIS data

MODIS measures the radiances from the earth's surface to acquire data in 36 spectral bands from 0.4 to 14.4 $\mu$m to provide information on the state of atmosphere, land, and ocean. Two bands of the 36 wavelengths image at a nominal resolution of 250 m, five other bands at 500 m, and the 29 remaining bands at 1 Km. The instrument achieves a 2330 Km viewing swarth and reaches near-global coverage every 1 to 2 days. MODIS, like MISR, provides different scientific data products from monitoring global dynamics and processes spanning the atmosphere, land, and the ocean. Each data products are available to the science community at different processed levels. This study uses cloud optical properties from collection 6.1, level-3 monthly aerosol-cloud dataset (MOD08) gridded at 1° x 1° horizontal resolution to present additional results. MODIS level-3 data is computed from the level-2 data statistics and inherits the data quality embedded in the corresponding level-2 products. For data quality assurance (QA), level-2 atmosphere products are processed through the initiation of validity flags algorithm (runtime QA flags) to better the data quality. These quality flags are broadly split into three parts, namely, cloud mask flags (include cloud mask status/cloudiness, day and night, sun glint, surface type flags), product quality flags (i.e., usefulness and confidence flags), and retrieval processing flags (atmospheric correction, Physical, algorithm, climatological constraints). The applicability of each sub-flags is a function of the parameter under consideration. More importantly, runtime QA flags are physically unavailable in Level 3 (L3) atmosphere products; instead, they are only used in computing (aggregating and weighting) L3 statistics and





apply to selected parameters. Therefore, selection of datasets by considering aggregation (e.g., day/night-time, liquid/ice clouds only, cloudy/clear, and surface type) and QA (mainly confidence level) in L3 data requires factoring in the characteristics of the study area. For this study, datasets including cloud optical thickness (COD), cloud fraction (CF), cloud top height (CTH),

liquid water path (LWP), cloud effective radius (CER), and atmospheric water vapour content (AWV) are used to derive more result. In MODIS retrieval, all the cloud properties are obtained using the infrared channel except for the AWV retrieved at the near-infrared band.

### 2.3  Meteorology data

Data obtained from SAWS stations were utilised to examine the impacts of aerosol and cloud on the meteorological condition

over the region. SAWS monitors and provides weather information services at several ground measurements stations across the nine provinces. The agency provides different ranges of data, from hourly to monthly data. The datasets used in this study include surface precipitation (PRECP), wind speed (WS), and ambient temperature (TEMP).

### 3  Results and discussion

### 3.1  Climatology of aerosol properties

Aerosol optical depth (AOD) still represents an important parameter in estimating the extent of aerosol loading over an area. Figure 2. shows the monthly averages for ten years (2007-2016) of MISR AOD measurement at 555 nm over South Africa. The adoption of AOD and other aerosol properties data from MISR instrument against that of MODIS lies on the higher grid resolution (0.5° x 0.5°) of the former compared to the latter, which is coarser (1° x 1°). Besides, previous studies demonstrated that MISR retrieved aerosol properties are in better agreement with ground-based instruments such as Aerosol Robotic Network

(AERONET) than MODIS (Abdou et al., 2005; Kahn et al., 2010; de Meij and Lelieveld, 2011; Tesfaye et al., 2011). As earlier pointed, the season over South Africa is typically similar to what is obtainable over the Southern hemisphere region such that summer begins in December to February, fall from March to May, winter from June through August, and spring from September to November. However, the meteorology characteristics are divided into two such that the upper and central parts mainly experience hot wet summer and cold, dry winter. Contrarily, the lower parts experience cold, dry summer, and cold, wet

winter, which influences the characteristics of aerosol emission over the different parts of the region. Also, the population and number of industrial activities in the upper part are more than the central, followed by the lower location. Subsequently, from Figure 2, AOD shows seasonality for the entire region. Hence, AOD increase in July through September (mainly during the spring months) followed by a general decline from October through June (mainly in winter). The maximum aerosol loading (AOD > 0.25) occurred during the September months and is primarily associated with biomass burning in the upper part of

the country and the surrounding bordering countries. Besides, aerosol emissions emerging from industrial and other human activities aggravate the aerosol particle suspension over the northern-most part of the region. Perhaps the dryness condition extending through the spring is instrumental in enhancing aerosol build-up, especially around Richard Bay (central east of the



country), where a coal loading bay and other industrial emissions occur. Aerosol particles removal by wet deposit is absent at this period of the year, thereby allowing these particles to remain in the air for a more extended period. Furthermore, there are

drifts of aerosol particles transported by air masses from the Northern to Southern part of the region. Most of these particles are fine mode aerosols such as aged smoke and dust particles of coal debris emitted into the atmosphere in the upper area and undergoes different interaction and complex reactions along the transit path. Meanwhile, the minimum AOD (typically < 0.05) is observed in June months and is mainly linked with aerosol removal process (e.g., dry and wet deposition, cloud scavenging) and less biomass burning activities. This result is similar to the observation of Formenti et al. (2002) and Tesfaye et al. (2011)

that high aerosol loading of mainly biomass burning and industrial smoke originating from Northern and border communities dominate aerosol loading over South Africa. Over the lower or southernmost part, biomass burning due to forest fire and industrial activity emissions also account for aerosol emission from November to March. However, due to the low turbidity, the effect is minimal compare to the central and upper parts. Furthermore, the region's central sub-region, especially around the west, contributes significantly to mineral dust (MD) suspension due to its arid nature. However, the impact is weaker compare

to other forms of aerosols over the region. The weakness could be due to the coarse nature of MD aerosol and the susceptibility to dry deposition. Although the process of aerosol removal or exit has not received much attention over South Africa (SA), there is a need for such study to give a better insight into the mechanism of aerosol emission, residence, and removal over the region. To complement the possible identification of biomass burning and general smoke emission observed from the AOD variation, Figure 3. illustrates the absorbing aerosol optical depth ($A_{abs}$). Like the AOD, the $A_{abs}$ demonstrates a seasonality pattern such

that an increase is observed from July to September then decreases from October through June. The maximum $A_{abs}$ (> 0.025) occurs in September and the minimum (< 0.005) during the January months. This variation indicates a concurrent increase in AOD and $A_{abs}$ during the spring season, thus demonstrating a significant increase in absorbing aerosol emissions such as biomass burning, fossil fuel combustion smokes, and carbon-related fine mode particles. Also, two essential characteristics of the $A_{abs}$ observation include revealing the period of high AOD associated with massive emission of absorbing aerosol particles

and the area dominating in concentration level. High absorbing properties appear more at the upper parts, then towards the central and lowest at the southernmost sub-region. Interestingly, a more significant portion of the absorbing trait is around the eastern parts than the west. Hence, considering the timing and location of peak values, biomass burning activities during the pre-farming period around Limpopo and bordering countries such as Botswana, Mozambique, and Zimbabwe contribute significantly (Formenti et al., 2002; Freiman and Piketh, 2003; Magi et al., 2009). The variation of Ångström exponent (AE)

in Figure 4. shows the predominance of fine mode aerosol in most periods of the year. An increase in fine mode aerosols from January to September, followed by a decrease during October to December, is evident over the region. The maximum AE value (AE > 1.7) is obtained chiefly during the August months, and the minimum (AE < 0.5) occurred in December. This observation is consistent with the changes in AOD and $A_{abs}$ in the previous figures. There are strong indications from both figures (Figure 2 and 3) that activities such as biomass burning, smoke emission from industries, and carbon-related dust

samples are responsible for high aerosol loading over the area. Besides, the regime of high aerosol loading exhibit seasonality pivoted around the summer and usually diffuse from the upper through the center then to the lower parts aided by the motion of air masses. Nevertheless, the lower and a fraction of the central sub-region are more of coarse mode and less absorbing





aerosol dominated areas. The characteristics of aerosol particles over this area earned their influence from proximity to the ocean, where sea salt aerosols are prevalent, and the arid region with a significant presence of MD, respectively. Furthermore,

some element of high presence of fine mode particles is noticeable around the coasts. Perhaps, these cases may be prompt by the occasional inflow of aged smoke over the Atlantic Ocean from the Amazon Forest to the west coast (see Formenti et al., 2002; Yakubu and Chetty, 2020) and the spread of aerosols to the Indian Ocean through the East coast.

## 3.2    Climatology of the atmospheric vapour and cloud optical properties

In this section, all parameters are based on MODIS instrument measurement and characterised by a horizontal resolution of

$1° \times 1°$ as earlier explained. The variation of atmospheric water vapour (AWV) or precipitable water is as shown in Figure 5. The observation of high precipitable water during summer is evident and low towards winter for all parts of the region. The highest vapour value (AWV > 3.5 cm) occurs during the January months and the lowest (AWV < 2.3 cm) in August. The evidence of seasonality in the pattern of atmospheric vapour is apparent here. Generally, AWV decline from February to August then rises from September through January. One important thing worthy of observation is that high atmospheric vapour

during the summer coincides with the precipitation season over the summer rainfall areas (mainly the upper and central parts of the region). Perhaps, there are chances of the high amount of vapour enhancing rainfall intensity over the region. Another important observation from the AWV variation is that it declines from north to south similarly to AOD, except for the coastal environments where the vapour content is relatively higher than the lower inland areas. The high value of vapour in summer in contrast to other seasons is due to warmer air conditions since warm air expands more and possesses the capabilities to

hold more water than cold air. A similar pattern is noticeable over the oceans around the region. AWV is higher over the Indian Ocean coast due to the warm air front from that direction than the lower value seen over the Atlantic Ocean, where cold wind emerges to the region. Earlier studies have demonstrated the seasonality of water vapour following the plunge to a minimum in June and rises to the maximum value in January (Yakubu and Chetty, 2020). Apart from the observation over this region, globally, AWV is found to increase in most areas during the summer, which is often associated with higher air

temperature (e.g., Stephens, 1990; Shie et al., 2006; Zhao, 2014). For the liquid water path (LWP) over South Africa, Figure 6 shows the distribution of cloud liquid content over the country. During winter months with extension to May and September, LWP is mainly low over the inland areas except for the coastal locations where relatively or higher values prevail. During this period, the values around the coasts reaches more than 40% higher than measurements obtained inland and most pronounced in June months. In contrast, LWP is more distributed over the entire region during summer, including November to April. The

maximum values (LWP > 20.0 $\mathrm{gm^{-2}}$) are witnessed during the June months and are mainly associated with the region's coasts. The minimum values (LWP < 10.0 $\mathrm{gm^{-2}}$) are predominant over the inland areas, particularly in August. Interestingly, both maximum and minimum occurred in the same seasonal period. Although the min-max happened during the same temporal domain seasonally, their spatial spaces are exclusively different. Meanwhile, the pattern demonstrated by LWP over this region does not show a discernible link with AWV especially considering the distribution of both parameters over the country. While

AWV is more distributed spatially over the central and upper sub-regions, LWP is more around the lower area. Nevertheless, carefully observing the variation of AWV and LWP, mainly during the summer and autumn seasons, gives further insight.



There is a sign AWV influences the changes in LWP, but other factors such as the aerosol types and perhaps temperature could be significant. For instance, there are periods of low AWV noticeably corresponding to relatively high LWP. Previous studies have identified less water vapour condensation resulting from less hygroscopic aerosols accountable for this feature

(Fan et al., 2012; Alizadeh-Choobari and Gharaylou, 2017). This observation is analogous to the spotted variation over South Africa, taking into account the high values of LWP over sea salt endowed lower parts compare to the inland (mainly central and upper) areas. In Figure 7, the variation of cloud optical depth (COD) is illustrated. Therein, COD is high mainly around the coastal areas compared to the inland environments. Besides, the eastern parts of the country domicile most optical thick clouds while the western parts of the region, except for the lower coastal parts, are more optical thinner clouds. Over the 10-years

study period, COD sparingly shows seasonality trends with the alternating increase and decrease in COD across the seasons. Higher values are seen mainly during summer and spring, then around the early parts of winter and fall. Meanwhile, the May, August, and September months are also characterised by the lower value of COD. The high COD value with extension to autumn corresponds to high atmospheric vapour and low aerosol turbidity during summer. In contrast, high COD during the spring season overlaps with the emission of a considerable amount of aerosol, predominantly fine mode particles. In the two

cases, there are possibilities that the high AWV and fine AOD significantly influenced the high COD, respectively. Meanwhile, the increase observed during winter might be accountable to cloud scavenging since both AWV and AOD are generally low at this period. Hence, there is a need for specific studies to understand aerosol removal processes over South Africa better. The maximum COD (> 22.0) occurred in June and the minimum (< 7.0) in May, just like the LWP. The high value of COD during summer and, to some extent, in autumn correspond to the period of high LWP and low AWV. In contrast, during the spring and

summer seasons, high COD coincide with high LWP and increasing AWV. In the two cases, there is evidence of robust affinity between the variation of LWP and COD. By mapping the idea of aerosols acting as nucleating seeds during cloud formation, the variation demonstrates the albedo effect proposed by Twomey (1977). However, the exception lies in the constant LWP in Twomey's postulation, which differs from the variation of this parameter as observed over the study region. Observations from other studies have also shown that COD increases with LWP, especially when there is a proportionate rise in $N_d$ (Wood, 2007;

Chen et al., 2015). Evaluation of the amount of cloud cover based on the cloud fraction in Figure 8 shows high value spreading from over the coastal environments towards the inland areas. Also, more cloud cover is observable around the upper eastern parts than the central and lower western parts except for the marine areas. Furthermore, high CF occurs during the summer and spring months and often declines towards autumn and winter, then rises through spring to summer again. The maximum values (CF > 0.7) were mainly recorded during the December months, while the minimums (CF < 0.25) were often around the August

periods. A vital outlook from the CF trend is that it increases with AWV. The fraction of cloud covered plays a crucial role in the amount of solar radiation reaching the surface, hence can significantly influence the thermodynamics of the clouds depending on cloud type. Large cloud cover often reduces the amount of solar radiation per unit area reaching the earth's surface. At the same time, lesser CF will enhance the quantity of incoming sun energy arriving on the earth. Although, there are other several important cloud properties such as the COD, the number of cloud droplets ($N_d$), and droplet size distribution that influences

the solar radiative energy over the earth. However, based on the surface area and relationship with cloud albedo, CF is an important factor defining the amount of sunlight reaching the ground (Liu et al., 2011; Muller et al., 2012; Smith et al., 2017).



Over South Africa, a lower temperature generally prevails during most of the year. The high-altitude feature of the region is one known factor that influences the low temperature. However, a rise in solar intensity is not uncommon over South Africa during the winter and spring months, which give rise to high temperatures during the day. During these periods, the low CF
over the region could enhance the temperature profile, thereby causing a slight warmth in the afternoon. Also, the eastern coast of the region is warmer than its western counterpart. The two air fronts from India (warm Agulhas current) and Atlantic Oceans (cold Benguela current) respectively are accountable for this and constitute the significant differences in climate and vegetation over the region (Wicking-Baird et al., 1997). As shown in Figure 9, the cloud effective radius (CER or $R_e$) is highest over the surrounding oceans and the coastal environment compared to the country's inland areas. Besides, $R_e$ is generally lower at the
upper parts of the country and increases navigating through the central location to the southernmost parts where values are maximum. Furthermore, high $R_e$ values are more spread over the western parts of the region than the eastern parts. In terms of temporal variation, CER basically increase during the autumn months and extends till winter. Then a declining pattern follows from the spring months till the end of summer in February. Also over the upper and central parts, $R_e$ is slightly higher than the lower location during the summer. Thus, the maximum values ($R_e > 15\ \mu$m) are mainly observed during the June months,
while the minimums ($R_e < 11\ \mu$m) are more during the July months. The high value of $R_e$ over the coastal area could be linked to the considerable presence of larger particle size aerosols, mostly of sea salt. In contrast, the lower value at the upper and central sub-regions, particularly towards the east, is associable with the dominance of fine mode particles. Several studies have elaborated on the role of cloud droplet size influencing the effective radius (e.g., Rosenfeld et al., 2006; Gustafson Jr et al., 2008; Fan et al., 2016). Hence, the dominance of Nd by large-sized particles will result in $R_e$ having high values, and small
particles accounting for the most $N_d$ will most result in low $R_e$. The variation played out over this region significantly aligned with the conventional theory by comparing the fine mode aerosol dominated upper and central parts with low CER values and the coarse mode associated lower parts with high CER. The distribution of cloud top height (CTH) over the region is illustrated in Figure 10. As expected, CTH is low over the ocean and to an extent at the coasts. Over the inland areas, CTH is more than two times greater than the height over the sea, depending on the period of the year. Temporally, CTH shows strong seasonality
such that elevation increase during the spring months through summer, then declines in autumn through winter. Also, cloud top height is highest over the upper, followed by the central and lower parts. The maximum value for the cloud top height (CTH > 7500 m) occurred during the summer months, especially in January, and the minimum (CTH < 4000 Km) in the winter period, mainly in the July months. A comparable variation is noticeable between AWV and CTH following their dramatic seasonal changes. Perhaps this characteristic could represent an increase in AWV, aiding more cloud development, especially at high
updraft velocity.

### 3.3   Climatology of aerosol-cloud interrelationship and meteorology characteristics

This section examines various aerosol and cloud properties observations based on the current understanding of aerosol-cloud interaction (ACI) and perhaps other distinct observable characteristics peculiar to the study region. The analysis attempts to enhance the knowledge of the influence of aerosol climatology on clouds and the reverse impacts (cloud effects on aerosol)
during different temporal periods. Further, this will provide an insight on the possible response of precipitation which is



valuable for future studies. The previous sections identified AOD as mainly high during the spring (August – November) and lowest in the winter season. To examine the interrelations amongst aerosol and cloud properties, Figure 11 shows the monthly mean variations of all parameters under consideration generated by statistical averaging of measurements over selected locations (Johannesburg 26.25° S, 28.05° E; Polokwane 23.90° S, 29.44° E; Nelspruit 25.47° S, 30.96° E; Mafikeng 22.81°

S, 25.50° E) in each of the provinces within the upper part. Critically scrutinising the three aerosol properties (AOD, $A_{abs}$ and AE), one can generally split the variation into two cases based on the combined AE-$A_{abs}$ relationship with AOD. The first case (case-I) represents the change in AOD with low AE < 1.0 with no significant influence of absorbing aerosol as seen during November to March. In comparison, the second case (case-II) represents the period (April-October) of changes in AOD associated with AE > 1.0 and corresponding variation in aerosol absorbing properties (AOD influenced by $A_{abs}$ increase). In

terms of the atmospheric vapour, case-I represents the relationship between AWV and AOD. From observation, AOD changes in the same manner (increase or decrease) as atmospheric vapour while the particle size is increasing/decreasing accordingly. This characteristic demonstrates a shift in the optical properties of aerosol particles (e.g., size and scattering albedo) due to water uptake. In the presence of high AWV, as particles with high affinity for water increase in size, AOD simultaneously increased, and AE decreases. The described pattern is especially true for an environment dominated by hygroscopic aerosol and

under humid conditions (Pilinis et al., 1989; Nair and Moorthy, 1998; Wu et al., 2018). Subsequently, the particles act as cloud condensation or ice nuclei (CCN or IN) and are later activated to cloud droplets at supersaturation by vapour condensation. The continuous and rapid growth of these cloud droplets explains the strong dependency of CF on AWV and AOD. In contrast, case-II illustrates AOD variation in response to rising absorbing particles such as organic carbon (OC) and black carbon (BC), where AWV is low. One should note the decreasing particle size (AE > 1.0) and sharp drop of CF with AWV. Coalescence

of cloud droplets formed reduces, compelling a proportionate reduction in the cloud cover (CF decreases). Besides, at the intersecting peak (in September) of the aerosol properties, a slight rise in vapour does not immediately increase particle size due to overwhelmed atmospheric water by fine mode aerosols. Cloud thickness has been explained to be influenced by LWP in the previous section. Hence, considering case-I, COD and LWP demonstrate strong dependency, suggesting proportionate cloud water to nucleating particle ratio. Contrarily, the dependence of the parameters (COD and LWP) is relatively weak

during case-II, such that the LWP does not directly quantify COD. Instead, cloud thickness increases with the aerosol particle size, and mostly the same for cloud droplets. Also, one can relate this to the variation of the CER in case-I, where COD is closely dependent on LWP correspond to a high effective radius ($R_e$ > 12 $\mu$m), hence large cloud droplet size and more cloud growth due to increase collision and coalescence. Furthermore, the increase in cloud top height along the effective radius and thickness will likely enhance radiative cooling at the top due to increasing cloud albedo. Interestingly, the surface

temperature is relatively high to invigorate a convective scene, prompting corresponding precipitation dynamics over the upper SA. Therefore, any shortfall in these parameters (CER, LWP, COD, CTH) to commiserate the aerosol concentration could be detrimental to precipitation occurrence. During case II, AOD is predominantly fine mode and mainly influenced by absorbing particles. Vapour is generally low, same as COD and LWP compared to the aerosol loading that can potentially act as CCN. This situation is portrayable as a polluted condition whereby aerosols nucleate a more significant amount of smaller cloud

droplets, thereby affecting the alignment of LWP and COD, resulting in lower CER and cloud cover. Consequently, cloud





droplets collision and coalescence rate are slower while cloud albedo increases according to the Twomey effect causing warm rain suppression (Albrecht, 1989; Ackerman et al., 2000). A vital observation is the radiative effect ushered by the smaller cloud droplets amid low surface temperatures. Radiative cooling at cloud-top along low surface temperatures enhances atmospheric stability and creates an unsuitable condition for convective developments (Yang et al., 2013). In Figure 12 representing the

variation over the central part made up of Bloemfontein (29.12° S, 26.22° E), Durban (29.92° S, 31.01° E) and Upington (28.39° S, 21.27° E). The dynamic is not much diverging from the upper part considering the drift of aerosol southward of the region and their sharing of similar meteorology conditions. Here, the aerosol-cloud interaction regime can be divided into two case studies as done for the upper part. Case-I covers the period (November to April) of high aerosol loading dominated by large size particles (i.e., AE < 1.0) and generally low absorbing aerosols. Meanwhile, case-II is the period (May to October)

where aerosol loading is fine particle dominated and linked with absorbing aerosol emission. Just as for upper SA, the vapour is higher during case-I, and CF maintains close variation as AWV. Cloud thickness significantly varies with cloud water following changes in AOD. A rise in AOD at high AWV corresponds to increasing COD, LWP, CER, CTH, and more precipitation. This process possibly follows the formation of larger size cloud droplets due to the concentration of bigger-sized particles yielding a proportionate increase in LWP and COD with a high value of CER and CTH, thereby leading to more precipitation. The

ambient temperature and wind speed are relatively high to drive the dynamic and thermodynamic aspects of the process. Noteworthy is the changes in precipitation rate due to characteristics features of aerosol-cloud interaction. For the second pattern (case-II) covering May to October, AOD shows substantial variation with absorbing aerosols and is primarily fine mode particle dominated. The meteorological condition is characterised by general low precipitation, temperature drop, and a slight reduction of horizontal wind speed. An increase in aerosol loading mainly corresponds to a decrease in COD and LWP.

Besides, CER generally decreases while CF and CTH decline in most parts of the period. On account of generally low AWV during case-II, CF and CTH decreased to a minimum but increased as AOD even in the presence of less water vapour around September. From the pattern, precipitation formation is motivated by relatively larger cloud droplet size (higher CER), high atmospheric water, and ambient temperature. COD and LWP increased and maintained proportionate variation, coupled with a slight increase in temperature, serving as potential invigorator of atmospheric instability, thereby raising the precipitation

chances. Over the lower part (Cape Town 33.92° S, 18.42° E and Port Elizabeth 33.95° S, 25.59° E) in Figure 13, the aerosol loading is generally low (AOD < 0.1) and further characterised by low cloud (CTH < 3 Km) typical of marine environments. Amazingly, this part presents high atmospheric vapour, ambient temperature, and wind speed during the summer and spring as the other parts (upper and central) yet low precipitation. Following a convention similar to the other parts, case-I represents the period between October and March where lower absorbing properties and AE ≤ 1.0 typify aerosol while rainfall is deficient.

In contrast, case-II during April – September demonstrates an increase in precipitation, absorbing aerosols and AE > 1.0 with AWV and temperature lower. During case-I, the general low aerosol loading corresponds to clouds with less water (low LWP) even though the cloud is optically thicker (COD > 15.0). There is notable contrast in the proportionate variation of LWP and COD coupled with low effective radius (CER < 14 $\mu$m). Precipitation decline here is linkable with low cloud water, droplet size, and reduced CER. The size of cloud droplet effective radius plays a vital role in precipitation forming clouds.

A low value in shallow clouds, particularly less than 14 $\mu$m, leads to slower collision and coalescence rates and precipitation





suppressing. Because more evaporation occurs in smaller size cloud droplets when mixing with dry ambient air, thus causing loss of cloud water. Several studies, including modelling experiments, have presented similar observations (Feingold et al., 2009; Chen et al., 2012; Fan et al., 2016). Meanwhile, case-II presented an increased aerosol loading period dominated by fine mode particles with strong absorbing properties where atmospheric vapour slightly dropped. Here, a rise in cloud water is

observable and proportionately varies with cloud depth while the ambient temperature dropped to minimal along with a slight decrease in wind speed. More importantly, CER is greater than 14 $\mu$m, and cloud height averagely increased sparingly. Thus, precipitation here is boosted by the increase in CER and LWP coupled with the low ambient temperature that will enhance the faster reach of supersaturation level. Because of the rise in CER, collision and coalescence will increase along cloud water since the coalescence rate is a function of droplet size.

**3.4 Interannual characteristics of aerosol, cloud, and meteorology parameters**

Similarities in the weather characteristics over central and upper South Africa constitute a vital outlook in the interannual variation in Figure 14. Critically observing the figure, the meteorological state of the region show close relationship amongst temperature (TEMP), wind speed (WS) and AWV, and a general increasing tendency from 2013. AOD, AE and $A_{abs}$ show a similar declining trend between 2012-2013, then increasing tendencies during 2008-2010 and 2013-2015 over the lower area.

Aerosol loading is generally low over this part (mostly AOD < 0.1), which is typical of a maritime environment (Smirnov et al., 2003). Even though a substantial amount of anthropogenic aerosols is emitted internally in this environment due to the high human population and industrial activities, long-range external emissions mainly trigger the aerosol loading and fine mode particles concentration (see Ichoku et al., 2003; Tesfaye et al., 2011). Cloud cover and top height over the lower sub-region experienced increments to two major peaks in 2009 and 2015, coinciding with the rise in AOD. An increase in aerosol loading

can enhance CF and CTH through the addition of CCN, invigorate existing cloud droplet concentration (Nd) to enhance vertical development (Niu and Li, 2012). The case is typical of warm clouds such as shallow cumuli and stratocumuli found over the marine environment as observed by previous studies (Pilinis et al., 1989; Koren et al., 2014). For atmospheric vapour, a distinct rise to the highest value in 2010 corresponds to high AOD, CF and CTH. Meanwhile, CER, LWP, and COD increased in 2008 and during 2012-2014, which fall into the years of moderate to highest rainfall and lower aerosol loading and AE. Likewise,

less precipitating years such as 2010 and 2015, mainly associated with drought events across the region (Yakubu and Chetty, 2020), coincide with high AOD and AE, while effective radius, cloud water, and optical thickness are lower. Over the central and upper sub-regions, AOD follows a series of increase and decrease almost along subsequent years. Nevertheless, AOD, AE, and $A_{abs}$ simultaneously rose to peaks in 2010 and 2015, just as water vapour. During these peaks, cloud cover is high over both upper and central parts except for the sharp drop at the upper area in 2015. Consequently, the growth in aerosol loading

over these sub-regions mainly enhanced by fine mode particles generally result in more cloud covers. In contrast, increasing aerosol loading largely negatively impacts cloud water and optical depth over the central sub-region, which resembles the trend over lower SA. However, the reverse is mostly the case over upper SA. The disparity in aerosol-cloud interaction behaviour might be due to differences in particle size distribution and the general aerosol composition over each area. As for the effective radius, an overall decrease prevailed during years associated with rising aerosol loading over the central and upper sub-regions.





This feature is more evident for 2010 and 2015 and consistent with observation over the lower sub-region. From the interannual

variations, several distinct characteristics of aerosol and cloud over each sub-region are noticeable apart from the dynamics of

aerosol-cloud interaction. For instance, the hierarchy by location of aerosol loading and optical depth of absorbing particles

demonstrate a declining trend from upper to lower. Meanwhile, for most cloud parameters, the reverse is the case such that an

increasing trend prevails from upper to lower area. Furthermore, parameters like CER, CTH, and CF are higher over the lower,

followed by the central and upper. These characteristics alone tend to set varying conditions for precipitation to occur.

### 3.5 Correlation analysis of parameters

From the previous sections, different patterns of aerosols and clouds properties in tandem with precipitation changes had been

observed to spread over the region. Notably, all the cloud properties have been observed to influence precipitation formation

differently over the geographical parts of the region. However, the impacts of CER and LWP show dominance among peers

over the entire study area. Meanwhile, the effects of other cloud properties (i.e., CF, CTH, and COD) are mainly pronounced

in the central and upper parts only. Conventionally, aerosols are observed to modulate clouds through aerosol-cloud interaction

(ACI), which affects precipitation depending on the feedback from ACI and other atmospheric circumstances. Studies have

demonstrated that an increment in CER and LWP often results in precipitation enhancement (?Fan et al., 2016; Freud and

Rosenfeld, 2012). Observation consistently found the increments to result from the rising collision and coalescence rates that

enhance cloud droplets size and precipitation formation (Chen et al., 2011; Fan et al., 2016). Numerous investigation and

numerical studies have often found CER to increase (decrease) with AOD depending on the size characteristics of aerosols.

Also, some studies have demonstrated the impact of rising aerosol loading on the decline in LWP and CER, thereby suppressing

rainfall (Albrecht, 1989; Jiang et al., 2009; Lebo et al., 2012). Meanwhile, CF and CTH rise owing to increasing aerosol loading,

although sometimes depending on cloud type (Chen et al., 2015; Christensen and Stephens, 2012). Irrespective of these various

findings, most of them are somewhat associated with considerable uncertainties, especially regarding the observations from

some other studies. To this end, more studies are required regionally and globally to understand the mechanism fully. This

section explores the relationships between aerosol and cloud, then cloud and precipitation to understand ACPI over South

Africa. Since particle size influences cloud properties (i.e., CCN and IN are predominantly fine mode aerosols), the correlation

analysis for aerosol properties considers AOD and aerosol index (AI). Figure 15a shows the correlation between AOD and cloud

properties (COD, LWP, CER, CF, and CTH). Generally, weak correlation coefficients (R) in the range of -0.4 < R < 0.24 exit

between and cloud properties over all parts of the region. In the case of the lower and central parts, all cloud parameters except

for CF and CTH (significance value P > 0.05) significantly related to AOD (i.e., P ≪ 0.05) despite their weak correlations (i.e.,

-0.4 < R < -0.20). Besides, their negativity regarding the correlation coefficient is consistent with earlier studies since aerosols

inhibit cloud development through the warming effect (Ten Hoeve et al., 2011; Jacobson, 2012). The low correlation might

result from aerosol types, atmospheric effects, and various uncertainties (e.g., aerosol and cloud retrieval and data coarseness).

Although, several observations and numerical evaluations have documented growth in both CF and CTH due to an increase

in aerosol loading (Fan et al., 2016). The cause of the insignificant and almost negligible R-value for CF and CTH for the

lower region and similar pattern for cloud cover in the upper parts is unclear and requires a close look. Further regarding the





upper sub-region, both CER (R = -0.22) and CTH (R = 0.23) are significant with P < 0.05. Again, the negative relation (i.e.,

AOD vs. CER) is consistent with the intuition that a high concentration of AOD results in more cloud droplets of smaller CER just as postulated by Twomey (1977). The positive relationship between AOD and CTH is explained by the cloud invigoration mechanism initiated through increasing aerosol loading, leading to vertical cloud development and increasing height (Chen et al., 2015; Christensen and Stephens, 2012). In Figure 15b, the relationship between AI cloud properties demonstrates a random correlation coefficient for each sub-region. For the lower part, only the relationship with COD seems significant

with R = -0.25 (weak correlation) and P « 0.05. The cause of the weak R-value is not clearly understood, mainly when no meaningful relationship is noticeable over other parts. However, COD is expected to be an essential radiative modulator of the cloud, which should be more inclined to the aerosol types. A previous study has also indicated no significant relationship between AI and COD (Costantino and Bréon, 2013). Over the central part, a much more significant but moderate correlation coefficient (R > 0.55) is observed for CF and CTH. This outcome suggests enhancing cloud cover and height with increasing

concentration of CCN and IN in the cloud. Observational studies have shown growing aerosol loading to enhance CF and CTH (Koren et al., 2010; Niu and Li, 2012). According to numeric simulation, this feature is further stressed to be enhanced by aerosol thermodynamic invigoration (Fan et al., 2013). Such occurrence is well observable in the transition from open to closed cells cloud. Results from satellite observation also demonstrate the increase in AI with stratocumulus clouds over the ocean (Gryspeerdt et al., 2014). On the contrary, the case is much different over the upper sub-region, with a mainly negative

correlation between AI and CF and CTH. The negativity is traceable to the changes in the thermodynamic properties of cloud by freshly induced aerosols since the upper part constitutes and is closer to primary sources of aerosols over the region. A positive correlation occurs between cloud properties and precipitation for the entire region, as illustrated in Figure 15c. The correlation coefficients range between 0.15 < R < 0.9 with ≈ 80% of the R-value representing moderate to strong correlation (i.e., 0.32 < R < 0.9). Nevertheless, weak relationships are observed for selected cloud properties (COD, CF, and CTH) in

the lower part of the region, with R typically < 1.8 for all properties. The weak correlation between rainfall and these cloud parameters at the lower sub-region lies mainly in the low aerosol loading and the general characteristics of shallow clouds. Notably, the significance of increasing cloud water and effective radius in enhancing precipitation is evident according to this chart as it cuts across the entire region. Hence, CER and LWP have demonstrated a critical role in deep and shallow clouds, as observed from this region. In general, based on the climatology of aerosol and cloud as described in the previous sections,

precipitation formation is a function of the characteristics of aerosol-cloud interaction. ACI leading to varying states of cloud properties, particularly CER, LWP, CF, and CTH, have differing effects on precipitation, hence, summarised in Table 1.

## 4    Conclusions

Air pollution through aerosol emission over the South Africa region consistently maintains a reoccurrence pattern centred around a sharp increase in spring. The bulk of these pollutants are transported to the area by air masses mostly originating from

the bordering countries. Domestic emissions mainly mix with transported incoming aerosols, hence, inducing a characteristic change in the atmospheric and environmental conditions. Nonetheless, the interaction between aerosol and the cloud is driven



by the dynamic and thermodynamic processes of the atmosphere, with aerosol and vapour acting as feedstocks. Even though aerosol-cloud interaction appears complex, the deduction from the analysis presented in this study shows that cloud formation and evolution is a function of the amount of aerosol and vapour integrated into the cloud system. Besides, the process itself
follows certain conditions that define the cloud properties and the ability to precipitate. Thus, infusing a proportionate concentration of aerosols and vapour into the cloud will enhance cloud water and effective radius (increase LWP and CER). Likewise, integrating a disproportionate amount of the duo will suppress rain-bearing cloud development because excess aerosol will result in smaller cloud droplets and lesser LWP. In contrast, extreme cloud water with fewer particles will surmount to the nucleation of fewer but bigger size droplets. Although, an increase in fine mode dominated aerosol corresponds to lower CER
and LWP, resulting in lower rain, as seen in this study. However, cases of predominated small size aerosol concentration leading to rising rainfall were also registered. Based on these assessments, the potential to precipitate is mainly the function of CER, LWP, and COD over the entire region. An increase in the three parameters increases the chances of rain-bearing cloud formation. Furthermore, the threshold for shallow lower subregion cloud for rain occur is CER > 14 $\mu$m, while the central and upper areas with more deep cloud possess a threshold of CER > 11 $\mu$m. Meanwhile, the average value for precipitating cloud
water over SA seems uniform for all the locations and corresponds to LWP > 100 gm$^{-2}$. Both CF and CTH influence more of the precipitation intensity. Both cloud cover and top height increase for high precipitating clouds and the reverse for low precipitating clouds. Nevertheless, the primary conditions (CER > threshold; LWP > threshold) for rain-bearing needed attainment before an increase in CF and CTH result in a pronounced rise in precipitation. So, an increase in cloud cover and depth will necessarily not yield rainfall but mainly will increase the quantity. Surface temperature and wind speed mainly increased
with atmospheric vapour. Due to the measurement altitude of both quantities (< 100 m above the ground), the influence is not much observed on aerosol loading and cloud development. Also important, temperature generally rises continuously in 2013 over the region.

*Data availability.* The aerosol and cloud data used in this work is freely available for download from the NASA ASDC (https://eosweb.larc.nasa.gov/) and LAADS (https://ladsweb.modaps.eosdis.nasa.gov/) websites, and the meteorology data is available through the SAWS website (https://www.weathersa.co.za/)
upon request.

*Author contributions.* ATY and NC designed the research. ATY processed and analysed the data, and wrote the manuscript. NC contributed to the write-up and participated in the review of the paper

*Competing interests.* The authors declare that they have no competing interest.





*Acknowledgements.* The authors gratefully acknowledge the NASA Langley Research Centre Atmospheric Science Data Centre (ASDC)
for providing the MISR and LAADS for the MODIS data. This authors also appreciates the support of the South Africa Weather Service
(SAWS) for making the meteorology data available.



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



**Table 1.** Table 1. A summary of the different states of cloud properties and the corresponding effects on precipitation over the region.

| CER/LWP | CF/CTH | PRECP | Characteristics |
|---------|--------|-------|-----------------|
| ↑ $TH$ | ↑ | ↑ | Moderate to heavy rain |
| ↑ $TH$ | ↓ | ↑ | Light to moderate rain |
| ↓ $TH$ | ↑ | ↓ | Cloudiness/non-drizzling to light rain |
| ↓ $TH$ | ↓ | ↓ | Clear sky/non-drizzling |

*Note: ↑TH and ↓TH respectively represent increment above and decrement below the threshold.





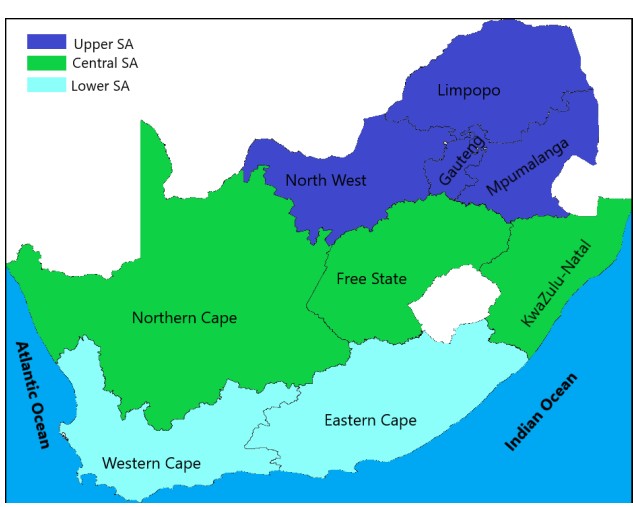

**Figure 1.** Map of South Africa divided into three sub-regions based on the distributions of aerosol loading.



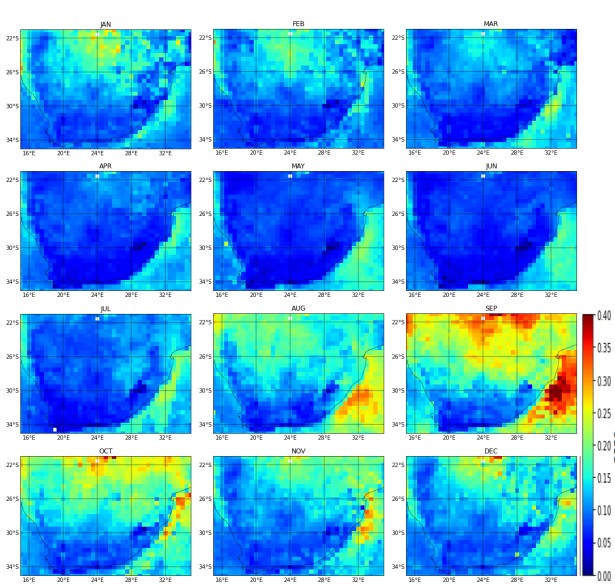

**Figure 2.** Average spatial distribution of aerosol optical depth (AOD) at λ = 555 nm over South Africa from MISR data for 10 years.

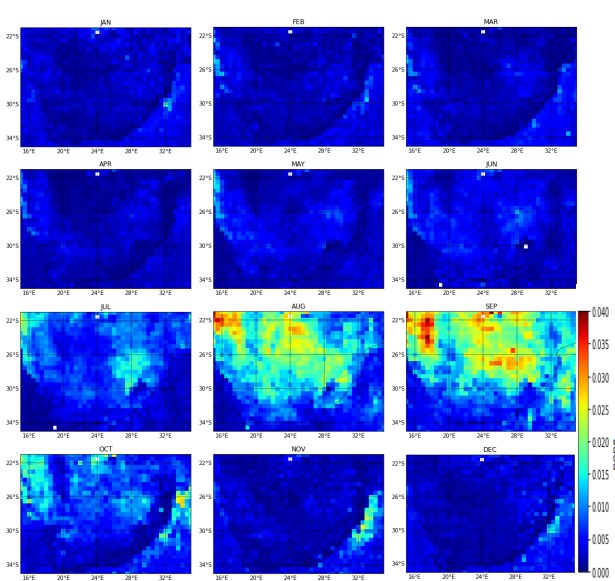

**Figure 3.** Average aerosol absorbing optical depth ($A_{abs}$) over South Africa from MISR data for 10 years.

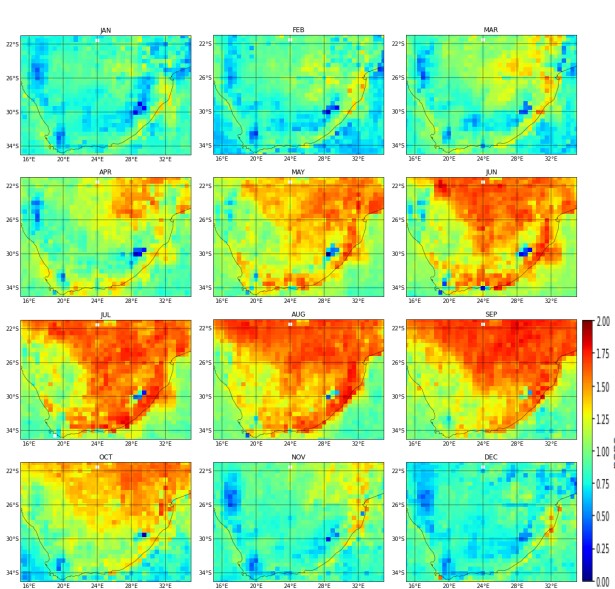

**Figure 4.** Average Ångström exponent (AE) over South Africa from MISR data for 10 years.





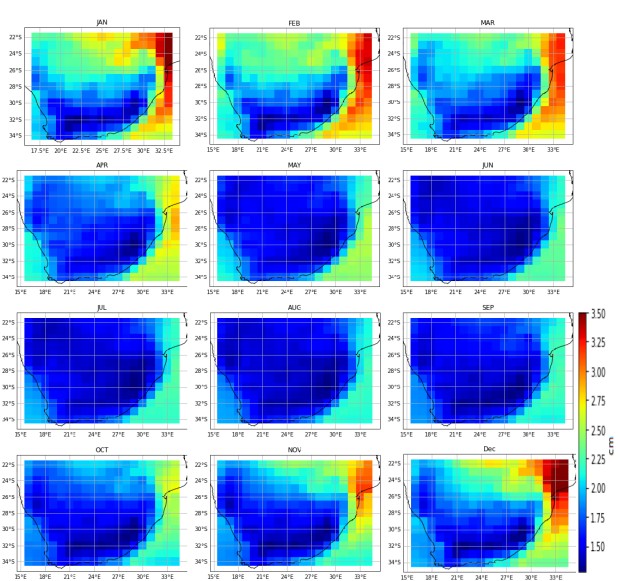

**Figure 5.** Average spatial distribution of atmospheric water vapour (AWV) in cm over South Africa from MODIS data for 10 years.

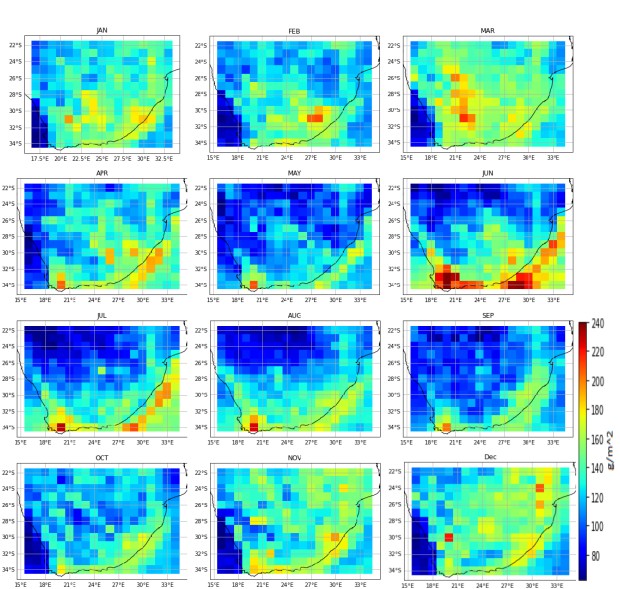

**Figure 6.** Average cloud water path (LWP) in gm$^{-2}$ over South Africa from MODIS data for 10 years.





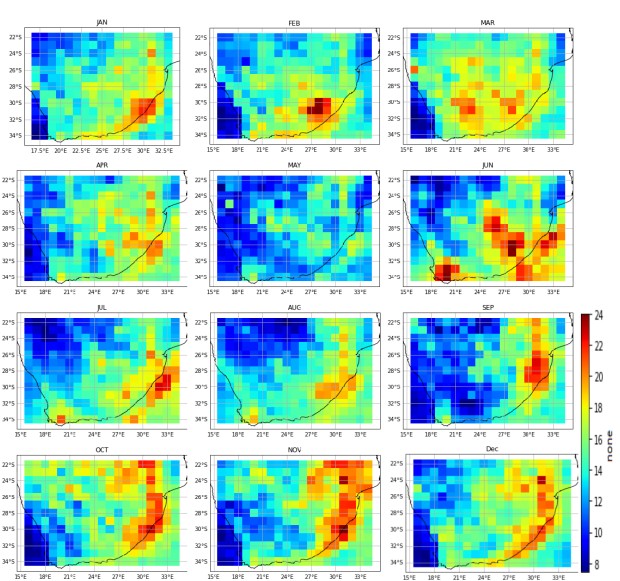

**Figure 7.** Average cloud optical depth (COD) over South Africa from MODIS data for 10 years.





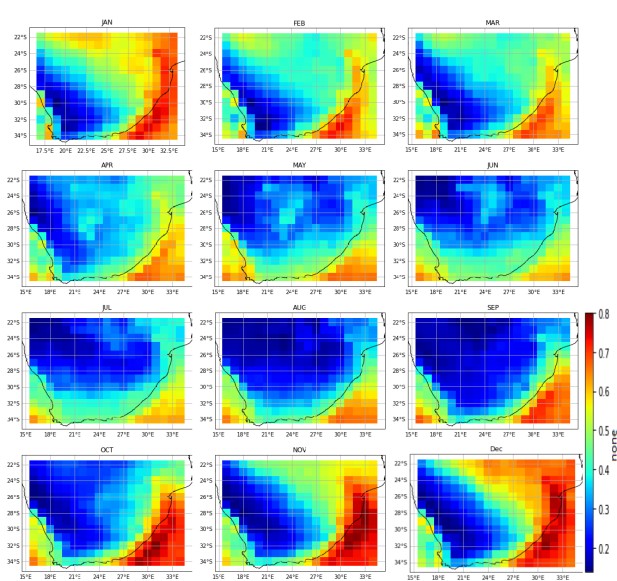

**Figure 8.** Average spatial distribution of cloud fraction (CF) over South Africa from MODIS data for 10 years.





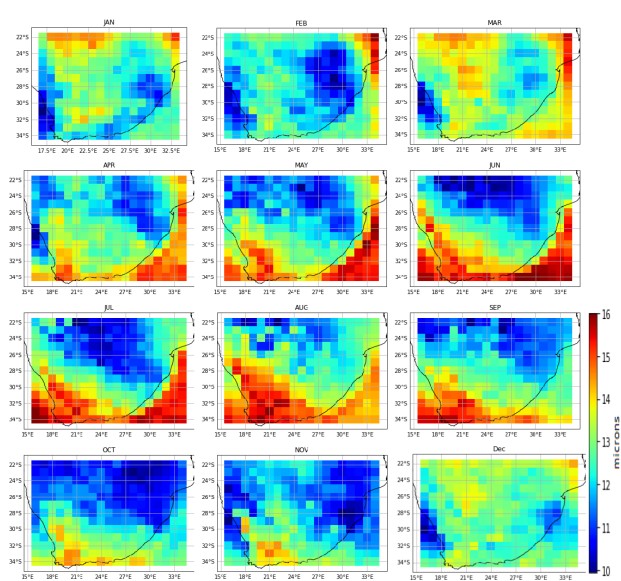

**Figure 9.** Average cloud effective radius (CER) in $\mu$m over South Africa from MODIS data for 10 years.



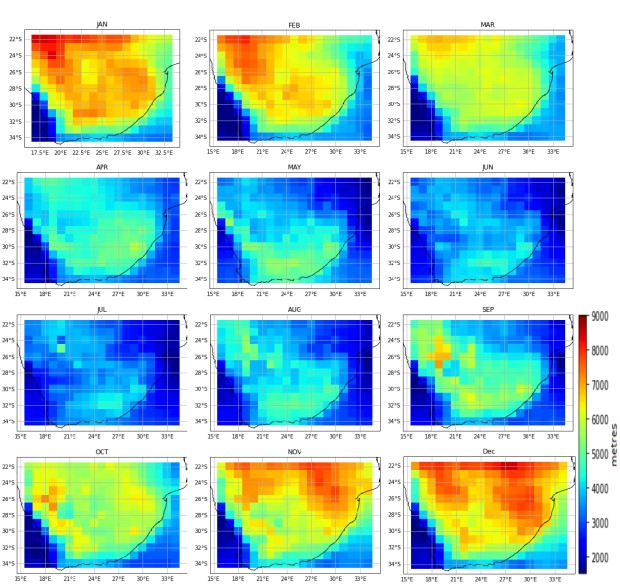

**Figure 10.** Average cloud top height (CTH)in m over South Africa from MODIS data for 10 years.



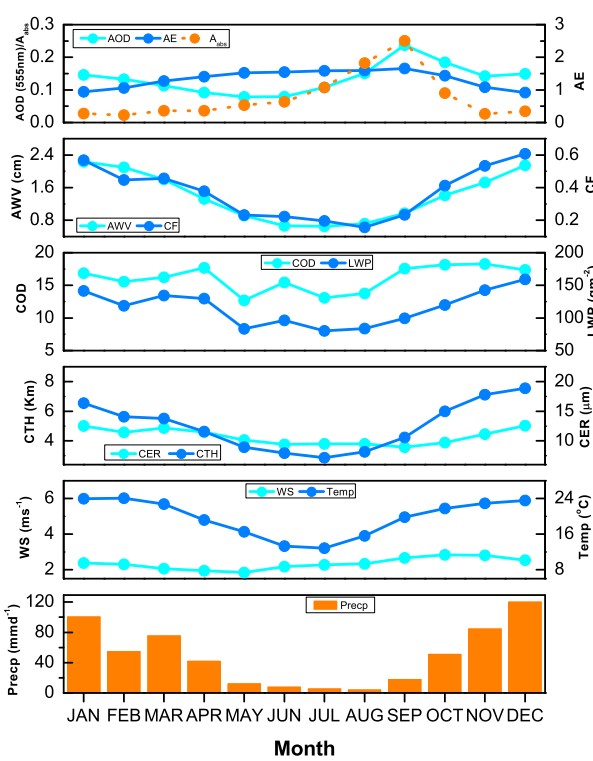

**Figure 11.** Intercomparison of average monthly variation of aerosol (AOD, AE, A$_{abs}$), AWV, cloud (CF, LWP, COD, CTH, CER) and meteorology (WS, TEMP, PRECP) parameters over upper South Africa.

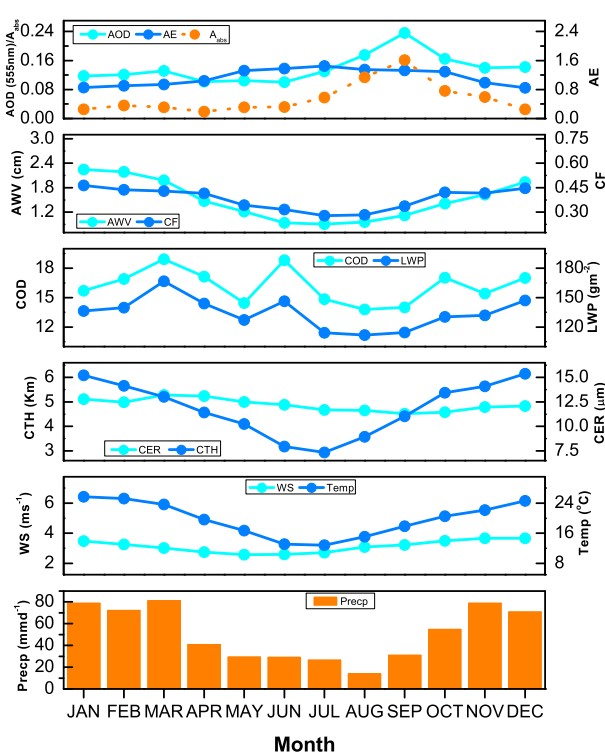

**Figure 12.** Intercomparison of average monthly variation of aerosol (AOD, AE, A$_{abs}$), AWV, cloud (CF, LWP, COD, CTH, CER) and meteorology (WS, TEMP, PRECP) parameters over central South Africa.

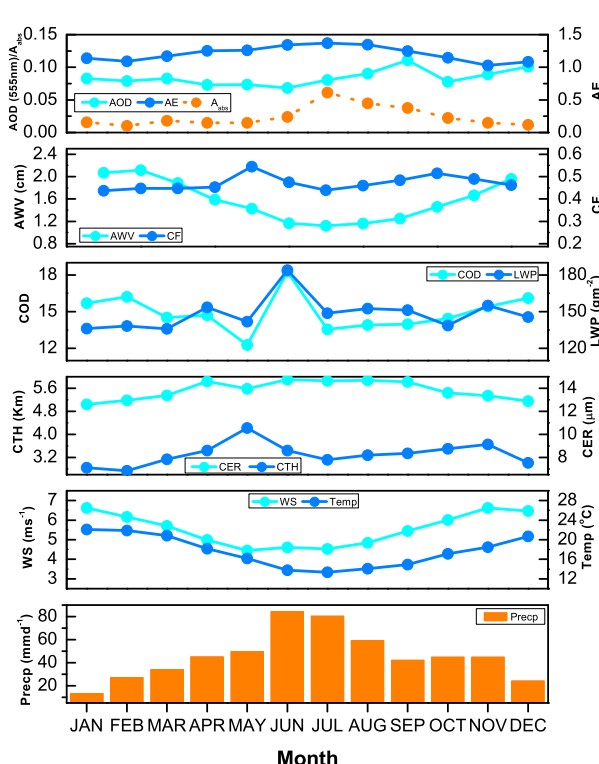

**Figure 13.** Intercomparison of average monthly variation of aerosol (AOD, AE, $A_{abs}$), AWV, cloud (CF, LWP, COD, CTH, CER) and meteorology (WS, TEMP, PRECP) parameters over lower South Africa.

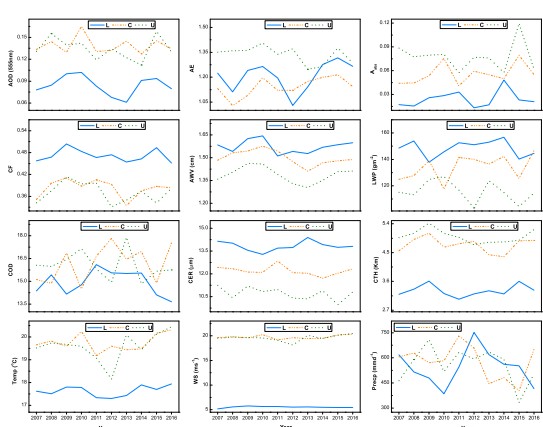

**Figure 14.** Interannual variation of monthly means for aerosol (AOD, AE, A$_{abs}$), AWV, cloud (CF, LWP, COD, CTH, CER) and meteorology (WS, TEMP, PRECP) parameters over upper (U; dotted green line), central (C; dotted orange line) and lower (L; solid blue line) South Africa.



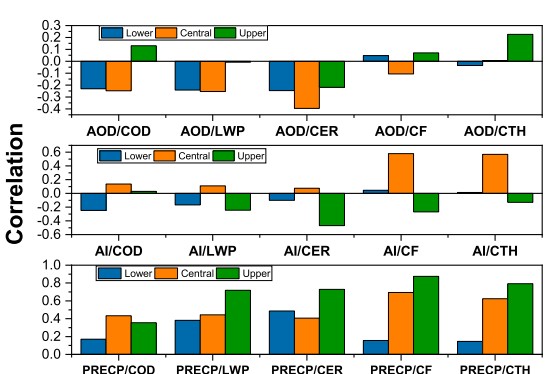

**Figure 15.** Correlation analysis of (a) AOD vs cloud properties, (b) AI vs cloud properties, and (c) PRECP vs cloud properties for upper, central and lower South Africa.