# Peer review of "A decadal assessment of the climatology of aerosol and cloud properties over South Africa"

_Atmospheric Chemistry and Physics, 2021_

## Author Comment (AC1)

**Major comments:**

*Meteorological data: Please give a better overview of how you derive precipitation, wind speed and temperature for the three regions. Is it mean values for all meteorological stations within one region? How many meteorological stations or hours of measurements are incorporated in this analysis? Or is it just data collected by the stations in the cities you mention later in the manuscript (Bloemfontein, Durban, Upington, Cape Town, Port Elizabeth, Johannesburg, Polokwane, Nelspruit, Mafikeng). If so, this should already be summarized in section 2.3 (e.g. by adding a table).*

An overview of the derivation of meteorological data over the region is provided in section 2.3 and summarised in a Table 1.

*Chapter 2.1. and Chapter 2.2 can be shortened: There is not really a need for such a very detailed description of the quality/algorithms/retrievals in MISR and MODIS products – if a reader is more interested on how the products are processed, a simple reference to the corresponding documents/webpages should be sufficient.*

Chapters 2.1 and 2.2 are now revised.

*The manuscript still has to be thoroughly revised regarding English grammar. In some parts of the manuscript I had a hard time to interpret what the authors want to describe.*

English grammar checked in entire manuscript

**Minor comments:**

*L18 and l23: Aerosols do not only interact with short-wave but also with long-wave radiation*

Statement on radiative effect due to aerosol influence on long-wave terrestrial radiation is now included in L18 and L23. Hence, the statements now read;

L18: "Aerosols interact with solar radiation directly through scattering and absorption, absorbs and re-emit long-wave radiation (LW) and indirectly relate with the cloud through cloud condensation nuclei (CCN) and ice nuclei (IN) formation to modify the earth's radiative budget"

L23: "Besides, they impact the reception and distribution of incoming solar and outgoing terrestrial (i.e., LW) radiations, thereby modifying the earth's energy budget"

L61: I don't understand the meaning of this sentence: 'Although climatology study is not holistic to diagnose the challenge of climate change, it identifies the next level of research that will clarify things for better understanding.'

The sentence is intended to emphasise the importance of climatology study in enhancing our understanding of climate change even though it cannot singly diagnose the phenomena.

For clarity, L61 rephrased as "Although climatology study cannot completely diagnose climate change, however, it is an important tool to identify significant trends that will help better understand the phenomenon."

L77: it is eight or nine years and not a 10-year period (2007-2015)

The study period is 10-years, hence,

L77: (2007-2015) >> (2007-2016)

L79: What means 'descending north to south poles'? I guess there must also be an ascending part of the orbit

The sentence intends to describe the path of the satellite as orbiting the earth from north to south pole.

L79: Hence, sentence rephrased as "... altitude of 705 km passing from north to south pole and overpasses the equator about 10:30 h ..."

L80: what is a 'swarth dimension' Is it the ground footprint of the swath (not swarth – recurring in the manuscript)

The correct phrase is "swath dimension" (i.e., the ground footprint of the swath). Therefore, "swarth" >> "swath" all where it appears in the manuscript (e.g., L80; 115)

L141: Here you state that the study is based on a period of 10 years (2007-2016) – however earlier you state that it is a period of 8 or 9 years (2007-2015) – please clarify

2007-2016 is the correct period as stated in the previous related comment

*L148 – L152: can you give numbers (e.g. climatological means of min/max temperatures, rain rate, population) for the different regions – this would help the reader and would specify the terms 'warm', 'cold', 'dry', 'wet' and 'higher population'*

Numbers illustrating the climatology characteristics and population of each sub-regions provided in Table 2.

L148-152: "See information on the meteorology characteristics and population of the sub-regions in Table 2.

*Figures 2-10: an outline of the defined regions (Upper, Central, Lower SA) on the map would make it much easier to interpret the findings.*

The regions represented as Upper, Central, and Lower SA on the maps (i.e., Figures 2-10) are already outlined in Figure 1.

*Chapter 3.1: you interpret the findings from MISR measurements by stating that biomass-burning aerosol is responsible for the peaks in AOD, A_abs, and AE. Your hypothesis would be substantiated if you would highlight regions with high activity of forest fires from spaceborne observations (e.g. VIIRS, MODIS) on the maps.*

To backup our hypothesis of biomass burning aerosol especially from the northern region and boundary countries is responsible for high cases of AOD, $A_{abs}$, and AE, Fig. S1 below shows the map of forest fire activity over the region as observed by Visible Infrared Imaging Radiometer Suite (VIIRS) for the peak months in 2013-2015. This figure is further provided as a supplementary material.

[Figure]

Fig. S1. A map of forest fire activity over South Africa and neighbouring countries as monitored by VIIRS instrument.

L249: N_d has not been introduced

$N_d$ now defined in L249 as "the number of cloud droplets ($N_d$)"

L262: what does 'lower temperature' exactly mean?

The phrase "lower temperature" is intended to describe the coolness of the region. The entire statement is revised as follow;

L262: "… a lower temperature generally prevails during …" >> "… coldness generally prevails during most of the year with temperature typically lower than 17 °C."

L464: Can the hypothesis that most of the pollutants are transported to South Africa from bordering countries be substantiated by literature? Otherwise it should be highlighted that this is just a hypothesis and not a fact – not to mislead the reader

Several references provided in the text have pointed to this observation (see, Formenti et al. 2002, Tesfaye et al. 2011, Yakubu and Chetty 2020). Please see also Fig. S1.

L413: citation: (?)

Omitted citation provided as "Rosenfeld et al., 2014"

**Technical corrections:**

L16: no brackets when citing IPCC

L26, you meant.  L26: (IPCC, 2013) >> IPCC (2013)

Throughout the whole manuscript: km (kilometer) not Km (Kelvin meter)

Km >> km in whole manuscript (e.g., L79; 81; 87; 115; 287; 356; Fig. 11-14)

L141: remove dot after 'Figure 2.' (recurring in the manuscript)

Dot after "Figure XX." (reoccurring in the manuscript) corrected as follow;

L141 "Figure 2." >> "Figure 2"

L174 "Figure 3." >> "Figure 3"

L185 "Figure 4." >> "Figure 4"

*Figures 2-4,7-8: The colorbars are labelled with 'none'. This might be confusing to the reader. Even though you show ratios, exponents and cloud fractions (which are without a unit), I would suggest to label the colorbar with the respective quantities (AOD, A_abs, CF, etc.)*

Colour bars for Figures 2-4; 7-8 labelled with their respective quantities

*L180 L236: 'than' not 'then'*

L180: "then" >> "than", and sentence now reads; "… at the upper parts than the central, and lowest at the …"

L236: "then" >> "than", and sentence revised as; "… spring than the early parts of winter …"

*L287: I guess it is CTH<4 km and not 4000 km*

(CTH < 4000Km) >> (CTH < 4km) now in Lxxx

*Figure 15 (a)-(c) are not indicated in the Figure*

Label (a)-(c) now indicated in Figure 15

*Some axes and colorbar labels are hard to read and should be enlarged*

Axes and colour bar labels for Figures 14-15 improved to ease readability

**Other modifications:**

L20: "… interaction and solar radiation …" >> "… interaction with solar radiation …"

L158: "… where a coal loading bay and other industrial emissions occur." >> "… where a coal loading terminal is located, and other industrial emissions occur."

L244: summer >> winter

L279 and L385: "Nd" >> $N_d$